# Validation of the Portuguese Version of the Fear of Progression Questionnaire-Short Form (FoP-Q-SF) in Portuguese Cancer Survivors

**DOI:** 10.3390/healthcare10122466

**Published:** 2022-12-07

**Authors:** Sandra Silva, Ana Bártolo, Isabel M. Santos, Débora Paiva, Sara Monteiro

**Affiliations:** 1Center for Health Technology and Services Research (CINTESIS), Department of Education and Psychology, University of Aveiro, 3810-193 Aveiro, Portugal; 2Center for Health Technology and Services Research (CINTESIS), Piaget Institute—ISEIT/Viseu, 3515-776 Viseu, Portugal; 3William James Center for Research, Department of Education and Psychology, University of Aveiro, 3810-193 Aveiro, Portugal; 4Department of Education and Psychology, University of Aveiro, 3810-193 Aveiro, Portugal; 5Department of Social Sciences and Management, University Aberta, 1269-001 Lisboa, Portugal

**Keywords:** fear of progression questionnaire, validation, cancer survivors

## Abstract

In 2020, around 60,000 people were diagnosed with cancer in Portugal, and many of them suffered some level of Fear of Progression (FoP) of the disease. Although this FoP is realistic, and is part of the normal and appropriate response to this type of disease, there is no instrument to assess and understand whether it is exaggerated in the face of the situation. The present study aimed to translate and validate the Fear of Progression Questionnaire-Short Form (FoP-Q-SF) for the Portuguese population. The sample consisted of 220 volunteers, aged 18 years or over and diagnosed with cancer for at least six months. Participants completed the FoP-Q-SF, the Hospital Anxiety and Depression Scale (HADS), the European Organization for Research and Treatment of Cancer Quality of Life Questionnaire Core-30 (EORTC QLQ-C30) and the Post-Traumatic Stress Disorder Checklist-Civilian Version (PCL-C). The FoP-Q-SF demonstrated high internal reliability (α = 0.86) and the confirmatory factor analysis supported the one-dimensional structure of the FoP-Q-SF. Convergent validity was supported with significant positive correlations with psychological distress, especially anxiety (0.68). The FoP-Q-SF has been found to be a valid instrument to measure FoP in Portuguese cancer survivors.

## 1. Introduction

Cancer is one of the leading causes of mortality in the world, with 18.1 million new cases of cancer and approximately 9.6 million deaths from the disease in 2018 [1]. In Portugal, the global trend is similar and there is a rapid and worrying growth in the number of new cancer cases per year, as well as an increase in the average age of the affected population [2]. In 2020, approximately 60,000 people were diagnosed with cancer in Portugal and of these, about 50% died [3]. Although the incidence of cancer continues to increase, in recent decades, advances in science and medicine have allowed for a decrease in the mortality rates and an increase in the survival rates of people diagnosed with cancer [4,5]. These people aim to live with dignity and quality, hence there is growing interest in the study of the side effects of cancer diagnosis and treatment on quality of life (QoL). Scientific research has shown that a cancer diagnosis is, in many cases, accompanied by distress (depression and/or anxiety) and that these are also inevitably associated with a lower QoL [6,7]. Between 10% and 30% of cancer patients present a clinical diagnosis of anxiety, which is a problem not only in the early stages of the disease but also in the long term. It should be noted that these figures are based on tools and criteria that have been developed for people with psychiatric anxiety disorders, such as adjustment disorder, generalized anxiety disorder, panic disorder, and phobic anxiety, and are generally not applied to people with chronic physical illness [8]. In the case of people with an oncological disease, their fears are not unreal and irrational; on the contrary, these patients are faced with a continuous and real threat, and they may develop concerns and fear that the disease will progress or recur, with all its biopsychosocial consequences. This fear is generally termed “Fear of Progression” (FoP) [9]. Although this FoP is realistic, and is part of the normal and appropriate response to this type of disease, the health professional responsible for the patient’s follow-up must be aware of situations in which this fear is exaggerated, considering the oncological diagnosis for which the patient is being treated or situations where the fear is present for a much longer period than would be expected. High levels of FoP can be dysfunctional and lead to a reduction in QoL. Therefore, adequate treatment and psychological intervention are necessary [9].

Since the FoP is distinguished from other anxiety disorders, to develop effective interventions, instruments that reliably assess the suffering of patients with this condition are needed. Herschbach et al. developed the Fear of Progression Questionnaire (FoP-Q) which showed good psychometric qualities in patients with chronic diseases (diabetes mellitus, rheumatic and inflammatory diseases, and cancer—Cronbach α = 0.70) [8]. The FoP-Q is a multidimensional self-report questionnaire consisting of 43 items evaluated on a five-point Likert scale, ranging from “never” to “very often”. The items are grouped into five dimensions: affective reactions, partnership/family issues, occupation, loss of autonomy, and coping with anxiety. In addition to this complete version, Mehnert et al. developed a short form, using a sample of patients with breast cancer residing in Germany [10]. This abbreviated version of the Fear of Progression Questionnaire (FoP-Q-SF) is composed of 12 items belonging to four of the five subscales (excluding coping with anxiety). The abbreviated form had a unidimensional factor structure, adequate reliability (Cronbach’s α = 0.87) and correlational analyses with other measures (Hospital Anxiety and Depression Scale (HADS), Post-Traumatic Stress Disorder Checklist-Civilian Version (PCL-C), Quality of Life Assessment Short Form-8 (SF-8), Life Attitude Profile-Revised (LAP-R)); therefore, it suggested adequate validity. The validation of the short version of the instrument was repeated in the German population, but with a wider sample that included patients with different types of cancer. The reliability of the FoP-Q-12 remained high (Cronbach’s α = 0.90) and exploratory factor analysis supported the one-dimensional structure, while confirmatory factor analysis only partially supported the one-dimensional model [11].

In addition to the German validations, the FoP-Q-SF was further translated to Mandarin and English and validated with a sample of cancer patients from Singapore. Both demonstrated high internal reliability (English: α = 0.87; Mandarin: α = 0.88) and test–retest reliability (English: r = 0.85, *p* < 0.01; Mandarin: r = 0.83, *p* < 0.01) [12]. The FoP-Q-SF was strongly correlated with other measures of FoP in cancer patients—the Fear of Cancer Recurrence Inventory (FCRI) (r = 0.66, *p* < 0.001) and the Fear of Recurrence Questionnaire (FRQ) (r = 0.64, *p* < 0.001)—and the factor structure was supported and replicated. More recently, this instrument was validated in Malaysia and showed excellent internal consistency (Cronbach’s α = 0.927) and good convergent and discriminant validity. Exploratory factor analysis of the FoP-Q-SF-M again supported the single-factor model reported by the English version of the FoP-Q-SF [13].

To date, there are no validated instruments for the Portuguese population that assess the fear of oncological disease progression. Considering the increase in the prevalence of cancer in Portugal, it is essential to develop instruments that seek to assess the variables that can influence not only the treatment but also the QoL of the person with this type of diagnosis. Thus, the present study aimed to translate and validate for the Portuguese population an instrument capable of assessing the FoP in cancer survivors.

## 2. Materials and Methods

### 2.1. Participants

In the present cross-sectional study, a convenience sample was recruited online, consisting of 220 volunteers of both sexes who met the following inclusion criteria: (i) aged 18 years or older and (ii) have had a diagnosis of oncological disease for at least six months.

### 2.2. Instruments

#### 2.2.1. Socio-Demographic and Clinical Questionnaire

In this investigation, a questionnaire, developed specifically for this purpose, was administered to collect relevant sociodemographic and clinical information, such as age, sex, education, professional status, marital status, the existence of children, clinical diagnosis, date of diagnosis, disease stage and phase, existence of metastases, treatments completed and/or in course, and psychological support.

#### 2.2.2. Fear of Progression Questionnaire-Short Form (FoP-Q-SF)

This instrument assesses the fear of chronic disease progression. Consisting of 12 items taken from four of the five subscales of the complete version by Herschbach et al. [8]: six items taken from the “Affective” subscale, two items from the “Occupation” subscale, two items from the “Relationship and Family” subscale and two items from the subscale “Loss of Autonomy”. Response to items is given on a Likert scale from 1 (“never”) to 5 (“very often”). The total score is calculated as the sum of the subscale scores, and varies between 12 and 60 points (see Appendix A); the higher the score, the greater the fear of disease progression. In its original version, this questionnaire has good psychometric properties, with a Cronbach’s alpha of 0.87 [10].

#### 2.2.3. Comparative Measures

##### Hospital Anxiety and Depression Scale (HADS)

The HADS was used to assess the severity of anxiety and depression symptoms among cancer survivors (Portuguese version by Pais-Ribeiro et al. [14]). It consists of two subscales, including seven items evaluating anxiety and seven items evaluating depression. Participants respond using a four-point Likert scale and each domain obtains a total score ranging from 0 to 21. Higher scores indicate higher level of symptomatology. The authors of the Portuguese version found a Cronbach’s alpha of 0.76 for anxiety and 0.81 for depression. We also found good internal reliability in the present sample (α HADS-A = 0.87, α HADS-D = 0.78).

##### The European Organization for Research and Treatment of Cancer Quality of Life Questionnaire Core-30 (EORTC QLQ-C30)

The EORTC QLQ-C30, validated by Pais-Ribeiro et al. [15] is a thirty-item tool developed to assess health-related QoL. This scale includes five functional scales, a global health status/QoL scale, three symptom scales, and single-item measures. In this study, only the following functional scales were administered: emotional functioning, physical functioning, role functioning and social functioning. Participants are invited to respond using a four-point Likert scale, ranging from “not at all” to “very much”. The scores for each subscale range from 0 to 100, with higher scores indicating better functioning. The internal consistency of the four subscales of the Portuguese version was high in the present sample, with the Cronbach’s alpha varying between 0.73 and 0.85 for role and emotional functioning, respectively.

##### Post-Traumatic Stress Disorder Checklist-Civilian Version (PCL-C)

The Portuguese version of this scale, which assesses the experience of a potentially traumatic event, was used [16]. The PCL-C has already been used in several contexts, namely with war veterans, victims of sexual harassment, victims of road accidents and work accidents, firefighters, police officers, the elderly, and people with cancer. Consisting of 17 questions that correspond to the 17 diagnostic criteria of post-traumatic stress disorder (PTSD) of the Diagnostic and Statistical Manual of Mental Disorders-IV-Text Revision (DSM-IV-TR) [17]. Responses are given on a five-point Likert scale, where 1 corresponds to “Not at all” and 5 corresponds to “Extremely”. Regarding the scoring, Marcelino and Gonçalves [16] indicate that it can be done in three ways: through the sum of the 17 items, with results varying between 17 and 85; through the sum of the items from the clusters “re-experience” (1–5), “avoidance” (6–12), and “hyperactivation” (13–17); or counting items with a score equal to or greater than 3. The Portuguese version revealed good psychometric characteristics, with a Cronbach’s alpha of 0.94. In the present sample, the Cronbach’s alpha is the same as in the Portuguese version (α = 0.94).

### 2.3. Procedure

To carry out the cross-cultural validation of the FoP-Q-SF instrument, authorization was requested from the author of the original version. The process of the translation into European Portuguese was performed involving the forward and backward (Portuguese–English) technique. Following the recommendations of Almeida and Freire [18] and Ribeiro [19], the translations and back translations were assessed by two expert psychologists specializing in psycho-oncology. The psychologists examined the different versions and discussed the items from the questionnaire to ensure a final version that was equivalent to the original in terms of the language used and adapted to aspects of culture. This process enables verification of comprehensibility of the questionnaire for Portuguese speakers. The final version was pre-tested with cancer patients. This phase allowed us to check the comprehensibility of the questionnaire for Portuguese speakers. This study was approved by the Data Protection Officer and by the Ethics and Deontology Committee of the University of Aveiro. The various instruments included in the research protocol were implemented in the LimeSurvey questionnaire platform of the University of Aveiro.

Subsequently, the access link to the online questionnaire was disseminated via email to several higher education institutions and to different institutions/organizations, such as the Portuguese League Against Cancer. In addition, the link was also disseminated through social media groups aimed at cancer patients on Facebook and Instagram. Data collection took place from October 2021 to January 2022.

### 2.4. Statistical Analysis

Statistical analysis was performed with the Statistical Package for Social Sciences (SPSS), version 28 (IBM Corp. Released 2021) and MPlus, version 6.12 (Muthén & Muthén, Los Angeles, USA). The sample characteristics were analyzed using descriptive statistics. One rule of thumb regarding sample size when performing a factor analysis is that the subject to item ratio should be at least 10 to 1 [18]. Based on this, the sample size was enough to ensure stability of a factor solution and, prior to testing factor structure of the FoP-Q-SF, data screening for normality and multicollinearity occurred. The factor validity of the scale was tested using confirmatory factor analysis (CFA). CFA models were estimated using weighted least squares with the mean and variance adjustment (WLSMV) estimator, a method recommended for ordered categorical data (e.g., Li [19]). The quality of the factor solutions was assessed through examination of the root mean square error of approximation (RMSEA), comparative fit index (CFI), Tucker–Lewis Index (TLI), and weighted root mean square residual (WRMR). A good-fitting CFA model should present a CFI ≥ 0.95, TLI ≥ 0.95, RMSEA ≤ 0.08 [20], and WRMR < 1.0 [21]. Convergent validity was assessed by examining Pearson correlations between the FoP-Q-SF and other validated scales that assessed related constructs.

## 3. Results

### 3.1. Sample Characteristics

Regarding sociodemographic characteristics, participants were aged between 18 and 79 years old (M = 48.80, SD = 9.43). Of the total number of participants (N = 220), 94.4% were female, 69.6% were married, 11.8% were single and 2.3% were widowed. Regarding schooling, 16.4% attended up to 9 years of schooling, 26.8% attended secondary education and 54.6% attended higher education. Regarding the composition of the household, most individuals lived with their spouse, children, and/or parents (83.7%) and only 12.7% lived alone. The professional status of the individuals at the time of data collection was distributed as follows: 46.4% were employed full-time, 1.8% were employed part-time, 3.2% were domestic workers, 9.5% were unemployed, 21.8% had temporary incapacity for work and 13.2% were retired.

The clinical characteristics of the sample participants are presented in Table 1.

### 3.2. Preliminary Analysis: Item Properties and Reliability

As shown in Table 2, all possible Likert scale answer values for each item were observed. The mean for most items was close to 3. The overall mean response for the 12 items was 40.4 *(SD* = 9.15). An assessment of normality showed that there were no deviations from the normal distribution, considering the absolute values of skewness and kurtosis [22]. All inter-item correlations were below 0.80, suggesting no multicollinearity. The corrected item–total correlations were all positive and larger than 0.30. A Cronbach alpha coefficient of 0.86 supported the good internal consistency of this measure, and there was a low variance in this coefficient if individual items were deleted. Based on this analysis, 12 items were retained for subsequent analyses.

### 3.3. Confirmatory Factor Analysis

To investigate the structural validity of the FoP-Q-SF, the original one-factor model [10] was estimated through CFA. The weighted least squares estimator was used due to the ordinal nature of the data. The results of the chi-square test were not considered, since research suggested that this analytical procedure is likely to reject well-fitting models when the sample size is relatively large [23]. All parameter estimates and their corresponding standard errors were statistically significant (*p* < 0.001). However, examination of the model fit indices indicated that the original unidimensional solution presented a poor fit to the observed data (CFI = 0.89; TLI = 0.87; RMSEA = 0.14; WRMR = 1.28). Similar to the study by Mahendran et al. [12], the modification indices suggested that the model fit could be improved if certain parameters in the error covariance matrix were freed. Thus, a modified one-dimensional model was tested, allowing the residual variance for the following pairs of items: item 1 and 2; item 4 and 12; item 4 and 5; and item 7 and 8. This revised model, conceptually justified, showed an acceptable fit (CFI = 0.96; TLI = 0.95; RMSEA = 0.08; WRMR = 0.8). The standardized factor loadings of the item parcels are presented in Figure 1.

### 3.4. Convergent Validity

The convergent validity was tested by correlating the FoP-Q-SF scores with the following scales: HADS depression, HADS anxiety, PTSD (re-experience, avoidance and hyperactivation scale), and QLQ-C30 quality of life (total, emotional, physical, role, social function). As expected, the strongest positive correlation was found between total FoP-Q-SF and anxiety scores, but FoP-Q-SF was also positively and moderately associated with depressive symptoms and post-traumatic stress. Although weaker, a positive correlation was also found between the FoP-Q-SF and the total score of the QLQ-C30 (see Table 3).

## 4. Discussion

Fear of disease progression (FoP) is one of the main causes of distress in cancer patients. This fear may or may not be exaggerated, but it is justified by a real threat and as such, is distinct from psychiatric anxiety. Specific instruments, which previously did not exist for the Portuguese population, are required to assess this construct. This study translated the items from the original English version of the FoP-Q into Portuguese, selected the 12 items belonging to the short version and investigated the reliability, factor structure and validity of the FoP-Q-SF in a Portuguese sample of cancer survivors.

The mean FoP-Q-SF score *(M* = 40.4) was higher than the scores reported in previous studies with this instrument [11,12,13]. These higher values may be due to the fact that the sample consisted almost entirely of women. It is known that women in the general population are more anxious than men, and that women with cancer are more anxious than male cancer patients [24]. Previous studies showed that the same is true for FoP [25].

The FoP-Q-SF demonstrated high internal reliability (α = 0.86), with coefficients greater than the cut-off criteria of 0.70 [18]. Our version revealed a Cronbach’s alpha close to the original version by Mehnert et al. [10] (α = 0.87) and the Mandarin version by Mahendran et al. [12] (α = 0.88).

Overall, the factor structure of the FoP-Q-SF was supported and replicated in the present study. The goodness-of-fit indices of the original model indicated some misfit, which could be adequately addressed if certain parameters in the error covariance matrix were freed. It is conceptually and statistically acceptable to allow residuals to correlate, as correlated errors are likely due to potential redundancy of item content [26,27]. In the present study, item 1 (“Fico ansioso se penso que a minha doença poderá progredir.”—“I become anxious if I think my disease may progress.”) and item 2 (“Fico nervoso antes das consultas médicas ou exames periódicos.”—“I am nervous prior to doctors’ appointments or periodic examinations.”), can be interpreted as very similar, with respect to general concerns associated with disease progression. Item 4 (“Tenho preocupações relativas ao alcance dos meus objetivos profissionais por causa da minha doença.”—“I have concerns about reaching my professional goals because of my illness.”) and item 12 (“O pensamento de que eu poderei não ser mais capaz de trabalhar devido à minha doença, perturba-me.”—“The thought that I might not be able to work due to my illness disturbs me.”) may also be perceived as similar concerns about occupational disruptions. Finally, item 7 (“Incomoda-me que possa ter que depender de estranhos para atividades do dia-a-dia”—“It disturbs me that I may have to rely on strangers for activities of daily living.”) and item 8 (“Estou preocupada que em algum momento da minha doença, eu não possa mais continuar com os meus hobbies.”—“I am worried that at some point in time, because of my illness I will no longer be able to pursue my hobbies.”) can also be perceived as similar concerns about the activities that constitute day-to-day life. As noted by previous validation studies on other FoP measures, these minor adjustments to improve model fit do not have any implications for the administration of the scale [26].

Different instruments were used to test convergent validity, namely the HADS, the PCL-C and the QLQ-C30. It was hypothesized that there were significant positive correlations between FoP and anxiety, depression and PTSD, and negative correlations with QoL. Convergent validity was supported with significant positive correlations with psychological distress, especially anxiety (0.68). This reveals that anxiety and FoP have a common characteristic: constant reflection on the future. Regarding the divergent validity, this was not confirmed. A negative correlation between FoP and QoL was expected, but a weak to moderate positive correlation was found. These results may be due to the fact that approximately 60% of the participants are in the remission phase and therefore their QoL is no longer as deteriorated as it was during the treatment phase. Future research should investigate how this is potentially subject to change with a sample with a more equal distribution of disease stages.

There were some limitations to be considered in this study. First, in this work, the direct and reverse translation method was used, which is very much based on exact translation of the text, reducing the freedom to adapt the concepts to the new language and population. This limitation may have been responsible for the apparent redundancy of some items. Second, the sociodemographic and clinical characteristics of the sample were not fully representative of the Portuguese cancer population. An example of this is the fact that approximately 55% of the sample is composed of people with higher education. Because it is an investigation utilizing online data collection and disseminated through social networks such as Facebook and Instagram, it may be more accessible to more literate people who will eventually make greater use of these tools. Third, as most participants were female and had breast cancer, we must be cautious in generalizing the study results. Finally, the cross-sectional design of this study, and the fact that most participants were in remission limit conclusions about the FoP course over time. More studies are needed to replicate and extend the current findings.

## 5. Conclusions

The FoP-Q-SF is the first valid, reliable, and clinically feasible questionnaire for Portuguese patients with an oncological disease. We believe that it can be used in future research and as a clinical tool to evaluate and monitor the FoP in this population.

## Figures and Tables

**Figure 1 healthcare-10-02466-f001:**
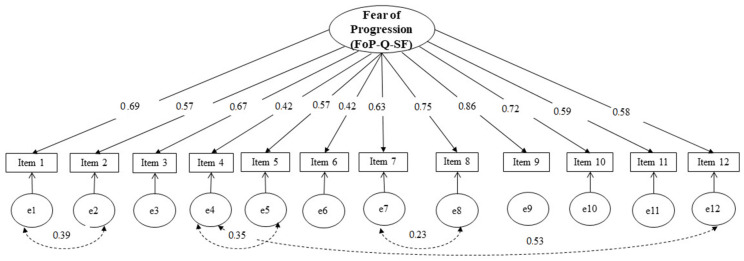
Modified one-factor model for the FoP-Q-SF.

**Table 1 healthcare-10-02466-t001:** Clinical characteristics of the sample (N = 220).

		N	%
Type of cancer	Breast	163	74.1%
	Lymphoma	7	3.2%
	Colorectal	6	2.7%
	Lung	5	2.3%
	Kidney	3	1.4%
	Stomach	2	0.9%
	Uterus	2	0.9%
	Leukemia	1	0.5%
	Melanoma	1	0.5%
	Pancreas	1	0.5%
	Prostate	1	0.5%
	Bladder	1	0.5%
	Sarcoma	1	0.5%
	Other	26	11.8%
Disease stage	I	23	10.5%
	II	43	19.5%
	III	76	34.5%
	IV	49	22.3%
	Unknown to the participant	29	13.2%
Disease phase	Diagnosis	2	0.9%
	Relapse	13	5.9%
	Remission	128	58.2%
	Treatment	77	35%
Time in months since diagnosis	≤12	35	15.9%
	>12 e ≤24	43	19.5%
	>24 e ≤36	39	17.7%
	>34 e ≤48	23	10.5%
	>46 e ≤60	22	9.6%
	>60	78	26.8%

**Table 2 healthcare-10-02466-t002:** Descriptive statistics of the FoP-Q-SF.

Item	M	Min	Max	SD	Skewness	Kurtosis	Corrected Item-Total Correlation	Cronbach’s Alpha If Item Deleted
1	3.40	1	5	0.981	0.063	−0.523	0.621	0.845
2	3.69	1	5	1.116	−0.379	−0.776	0.497	0.841
3	3.20	1	5	1.172	0.078	−0.913	0.554	0.847
4	2.97	1	5	1.407	−0.021	−1.278	0.438	0.857
5	3.40	1	5	1.066	−0.273	−0.490	0.512	0.850
6	3.25	1	5	1.435	−0.279	−1.191	0.334	0.865
7	3.38	1	5	1.286	−0.301	−1.026	0.553	0.847
8	3.27	1	5	1.133	−0.087	−0.753	0.684	0.839
9	3.25	1	5	1.211	0.007	−1.011	0.870	0.837
10	3.46	1	5	1.280	−0.370	−0.860	0.592	0.845
11	3.85	1	5	1.087	−0.571	−0.552	0.503	0.851
12	3.27	1	5	1.332	−0.248	−1.033	0.553	0.848

**Table 3 healthcare-10-02466-t003:** Convergent validity: correlations between Fear of Progression (FoP-Q-SF) and Distress (HADS-D and HADS-A), quality of life sub-dimensions and post-traumatic stress disorders.

Questionnaire Scales	Mean Total Score	FoP-Q-SF
HADS—Anxiety subscale	8.79	0.687 **
HADS—Depression subscale	6.04	0.480 **
PSPT—Total Score	2.75	0.507 **
PSPT—Reexperience subscale	2.73	0.464 **
PSPT—Avoidance subscale	2.60	0.424 **
PSPT—Hyperactivation subscale	2.96	0.488 **
QLQ—Total Score	39.38	0.380 **
QLQ—Emotional functioning subscale	49.37	0.576 **
QLQ—Physical functioning subscale	70.68	0.470 **
QLQ—Role functioning subscale	62.00	0.386 **
QLQ—Social functioning subscale	49.37	0.541 **
FoP-Q-SF—Total Score	40.40	_

** *p* < 0.01.

## Data Availability

The data that support the findings of this study are available from the corresponding author, [S.S.], upon reasonable request.

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
