# Peer review of "Validation of the Portuguese Version of the Fear of Progression Questionnaire-Short Form (FoP-Q-SF) in Portuguese Cancer Survivors"

_healthcare, 2022, doi:10.3390/healthcare10122466_

Round 1
Reviewer 1 Report
1. Good and scientifically sound study , other similar studies were done before as well [e.g. 1. Fear of progression in breast cancer------fear of progression questionnaire (FoP -Q-SF) . Anja Mehnert et al .Z Psychosom Med Psychother 2006 ;52 (3): 274-88) ,2. Validation of the English ------short form in Chinese cancer survivors . BMC Psychology 2020 ;8 (10)] , but restricted to particular cancer and population other than Portuguese.
2. Inclusion criteria should also mention Sex (pregnant and lactating women) . Exclusion criteria is missing (include if any)
3. Sample size calculation method and source not mentioned
4. Some more recent references may be included
5. Validation and source of Socio-demographic & Clinical Questionnaire -not mentioned
Author Response
We appreciate the attention given to our manuscript. The suggestions given contributed to the improvement of this work. We hope we understood all the suggestions and made the necessary changes.
Below are the changes made so they can be more easily identified in the manuscript.
- Inclusion criteria were included. Exclusion criteria did not exist.
- Sample size calculation method and source were added in section 2.4.
- The sociodemographic and clinical questionnaire was not validated. It was an instrument built by the authors, with the sole purpose of obtaining personal and clinical data.
Reviewer 2 Report
This is a well written and interesting paper about a relatively new concept that is gaining increasing importance in the field of health psychology. The authors created a Portuguese Version of a relatively new scale that was developed to quantify the "fear of progression" in cancer survivors. Employing a "convenience sample" of 220 online subjects, whose general education appeared to exceed that of the average individual, the authors demonstrated that the short form of the Portuguese Fear of Progression Questionnaire demonstrated good internal consistency and correlated positively at highly significant levels with other similar measures. They conclude that the Portuguese version of the the test can be used confidently to assess fear of progression of cancer in their country. I had a bit of trouble with the comment starting on line 269: "In the present study, Item 1...etc". The translations of items 1 and 2 in English are identical but they not identical in Portuguese. That is going to appear very strange to most readers. Is it possible to show the nuanced differences between the two items in English?
Author Response
We appreciate the attention given to our manuscript. The suggestions given contributed to the improvement of this work.
There was a writing error in items 1 and 2 that has been corrected.
Reviewer 3 Report
The authors present an interesting and important issue. The translation and validation of the Portuguese Version of the Fear of Progression Questionnaire-Short Form is methodologically well done. Below are some comments and suggestions for alterations.
1. Abstract:
The wording of “proved to be” could be changed, maybe “has been found to be” would be more appropriate.
2. Introduction:
Good overview of the thematic background of fear of progression and recurrence of the disease and relevance of assessing it. Other language versions of the questionnaire that have already been adapted and validated are presented well. One minor point for potential alteration is the wording of “survival years” (page one) which brings a negative connotation with it, especially in cases where cancer patients are young.
3. Materials and Methods:
Overall very clear and containing all necessary information. Note that sometimes the numbers are written out and other times in digits (e.g. 6 months on page three). Consistency could support a better reading flow and is more appropriate for a scientific article.
One major point in the procedure section (2.3 on page 4): The forward and backward translation technique has certain weaknesses (such as being very text-based and thus not allowing for some degree of freedom that might make a better fit in the new language) which should be discussed. The current best practice approach is the TRAPD-method by Harkness (2003) which is a team / committee-based approach. Having the translation done by two expert psychologists specializing in psycho-oncology is great. Additionally a linguist or language expert would be a very valuable addition for the translation process.
4. Results:
The statistical steps are well described and show good methodological practice. Some minor alterations should be made for the mean, standard deviation etc. to be written in italics. In table 3, the meaning of the indication for the p-values (*) should be explained below or in text.
5. Discussion:
The section is well written. It would be good if the authors reconsidered the constructs of "irrational anxiety" and "psychiatric anxiety" in comparison with "fear of disease progression". I don't think that the two can be distinguished in the sense that progression anxiety is always well-founded and real. fear of disease progression can also lack a real basis, and patients with the same stage of disease, e.g., with a very good prognosis, can be affected very differently. With this in mind, please amend the first paragraph of the discussion once again.
A good feature is the critical discussion of the higher total mean score for fear of progression and a potential gender effect. The process of reaching a better model fit by allowing residuals to correlate seems appropriate. However, if therefore there is an indication of potential redundancy in item content, you may have considered to exclude some items for better fit and also to make the short scale even less time-consuming and more easy to work with from a patient perspective.
Item one and two in English are the same, which seems to be an error. Item two, according to Herschbach, should be: I am nervous prior to doctor’ appointments or periodic examination. Check the numbering and English versions of the items.
The lack of confirmation of divergent validity is explained by high quality of life and also high fear of progression due to 60% of participants being in the remission phase, which seems plausible. Future research should investigate how this is potentially subject to change with a sample with a more equal spread of phases of the disease.
In the last paragraph of the discussion section, you may want to consider whether Facebook and Instagram as platforms will achieve a representative sample. In addition, under limitations the following is mentioned: “A higher education level may be associated with greater health literacy and, consequently, greater FoP”. Evidence for this is missing and should be added.
Author Response
We appreciate the attention given to our manuscript. The suggestions given contributed to the improvement of this work. We hope we understood all the suggestions and made the necessary changes.
Below are the changes made so they can be more easily identified in the manuscript.
Abstract
-The suggestion was followed and the wording "proved to be" was changed to " has been found to be”.
Introduction
-The sentence was changed to eliminate “survival years”.
Materials and methods
-The newspaper's directions followed, the numbers from zero to nine were written in full and the remaining numbers in digits.
Results
-The journal instructions do not indicate that the means and standard deviations should be written in italics. Other journal publications were reviewed, and the mean and standard deviation were not italicized.
-In table 3, the meaning of the indication for the p-values (*) was added.
Discussion
-The fear of progression always has a real physical threat, although it can be more or less exaggerated. Unlike psychiatric anxiety, in which it threatens, it is often not present. The authors of the original instrument themselves make this distinction in the following publication: Herschbach, P.; Berg, P.; Dankert, A.; Duran, G.; Engst-Hastreiter, U.; Waadt, S.; Keller, M.; Ukat, R.; Henrich, G. Fear of Progression in Chronic Diseases. Journal of Psychosomatic Research 2005, 58, 505–511, doi:10.1016/j.jpsychores.2005.02.007.
-In fact, in items 1 and 2 there was a translation error that was corrected.
-Suggestion for future research has been included.
-The last paragraph of the discussion has been changed as suggested.
Round 2
Reviewer 3 Report
Overall the suggested changes were made. A few minor points regarding the font in the results section should be considered:
· Number of table / figure in bold (e.g. Table 1, then description in italics as already done correctly in the manuscript)
· Mean, Standard Deviation etc. in italics
In the discussion section, the suggested remark whether FoP is always rational and justified was added. Note the incorrect word order (“threat real” on page 7). Also, one sentence about the limitations of the forward-backward-translation approach would be a good addition to the critical evaluation.
If these points are integrated into the revised manuscript, it is fit for publication.
Author Response
Once again thank you for your precious contribution. I made the suggested changes. I had doubts about what was requested to change in the title of the tables.However, I did it according to the newspaper's template. This is "table 1" in bold and the rest of the text in normal font.
